# Hard Shape-Constrained Kernel Machines

**Pierre-Cyril Aubin-Frankowski**
École des Ponts ParisTech and CAS
MINES ParisTech, PSL
Paris, 75006, France
pierre-cyril.aubin@mines-paristech.fr

**Zoltán Szabó**
Center of Applied Mathematics, CNRS
École Polytechnique, Institut Polytechnique de Paris
Route de Saclay, Palaiseau, 91128, France
zoltan.szabo@polytechnique.edu

## Abstract

Shape constraints (such as non-negativity, monotonicity, convexity) play a central role in a large number of applications, as they usually improve performance for small sample size and help interpretability. However enforcing these shape requirements in a *hard* fashion is an extremely challenging problem. Classically, this task is tackled (i) in a soft way (without out-of-sample guarantees), (ii) by specialized transformation of the variables on a case-by-case basis, or (iii) by using highly restricted function classes, such as polynomials or polynomial splines. In this paper, we prove that hard affine shape constraints on function derivatives can be encoded in kernel machines which represent one of the most flexible and powerful tools in machine learning and statistics. Particularly, we present a tightened second-order cone constrained reformulation, that can be readily implemented in convex solvers. We prove performance guarantees on the solution, and demonstrate the efficiency of the approach in joint quantile regression with applications to economics and to the analysis of aircraft trajectories, among others.

## 1   Introduction

Shape constraints (such as non-negativity, monotonicity, convexity) are omnipresent in data science with numerous successful applications in statistics, economics, biology, finance, game theory, reinforcement learning and control problems. For example, in biology, monotone regression techniques have been applied to identify genome interactions (Luss et al., 2012), and in dose-response studies (Hu et al., 2005). Economic theory dictates that utility functions are increasing and concave (Matzkin, 1991), demand functions of normal goods are downward sloping (Lewbel, 2010; Blundell et al., 2012), production functions are concave (Varian, 1984) or S-shaped (Yagi et al., 2020). Moreover cyclic monotonicity has recently turned out to be beneficial in panel multinomial choice problems (Shi et al., 2018), and most link functions used in a single index model are monotone (Li and Racine, 2007; Chen and Samworth, 2016; Balabdaoui et al., 2019). Meanwhile, supermodularity is a common assumption in supply chain models, stochastic multi-period inventory problems, pricing models and game theory (Topkis, 1998; Simchi-Levi et al., 2014). In finance, European and American call option prices are convex and monotone in the underlying stock price and increasing in volatility (Aït-Sahalia and Duarte, 2003). In statistics, the conditional quantile function is increasing w.r.t. the quantile level. In reinforcement learning and in stochastic optimization the value functions are regularly supposed to be convex (Keshavarz et al., 2011; Shapiro et al., 2014). More examples can be found in recent

surveys on shape-constrained regression (Johnson and Jiang, 2018; Guntuboyina and Sen, 2018; Chetverikov et al., 2018).

Leveraging prior knowledge expressed in terms of shape structures has several practical benefits: the resulting techniques allow for estimation with smaller sample size and the imposed shape constraints help interpretability. Despite the numerous practical advantages, the construction of shape-constrained estimators can be quite challenging. Existing techniques typically impose the shape constraints (i) in a 'soft' fashion as a regularizer or at finite many points (Delecroix et al., 1996; Blundell et al., 2012; Aybat and Wang, 2014; Wu et al., 2015; Takeuchi et al., 2006; Sangnier et al., 2016; Chen and Samworth, 2016; Agrell, 2019; Mazumder et al., 2019; Koppel et al., 2019; Han et al., 2019; Yagi et al., 2020), (ii) through constraint-specific transformations of the variables such as quadratic reparameterization (Flaxman et al., 2017), positive semi-definite quadratic forms (Bagnell and Farahmand, 2015), or integrated exponential functions (Wu and Sickles, 2018), or (iii) they make use of highly restricted functions classes such as classical polynomials (Hall, 2018) or polynomial splines (Turlach, 2005; Papp and Alizadeh, 2014; Pya and Wood, 2015; Wu and Sickles, 2018; Meyer, 2018; Koppel et al., 2019). Both the polynomial and spline-based shape-constrained techniques rely heavily on the underlying algebraic structure of these bases, without direct extension to more general function families.

From a statistical viewpoint, the main focus has been on the design of estimators with uniform guarantees (Horowitz and Lee, 2017; Freyberger and Reeves, 2018). Several existing methods have been analyzed from this perspective and were shown to be (uniformly) consistent, on a case-by-case basis and when handling specific shape constraints (Wu et al., 2015; Chen and Samworth, 2016; Han and Wellner, 2016; Mazumder et al., 2019; Koppel et al., 2019; Han et al., 2019; Yagi et al., 2020). While these asymptotic results are of significant theoretical interest, applying shape priors is generally beneficial in the small sample regime. In this paper we propose a flexible **optimization framework** allowing multiple shape constraints to be jointly handled in a hard fashion. In addition, to address the bottlenecks of restricted shape priors and function families, we consider general affine constraints on derivatives, and use reproducing kernel Hilbert spaces (RKHS) as hypothesis space.

RKHSs (also called abstract splines; Aronszajn, 1950; Wahba, 1990; Berlinet and Thomas-Agnan, 2004; Wang, 2011) increase significantly the richness and modelling power of classical polynomial splines. Indeed, the resulting function family can be rich enough for instance (i) to encode probability distributions without loss of information (Fukumizu et al., 2008; Sriperumbudur et al., 2010), (ii) to characterize statistical independence of random variables (Bach and Jordan, 2002; Szabó and Sriperumbudur, 2018), or (iii) to approximate various function classes arbitrarily well (Steinwart, 2001; Micchelli et al., 2006; Carmeli et al., 2010; Sriperumbudur et al., 2011; Simon-Gabriel and Schölkopf, 2018), including the space of bounded continuous functions. An excellent overview on kernels and RKHSs is given by Hofmann et al. (2008); Steinwart and Christmann (2008); Saitoh and Sawano (2016).

In this paper we incorporate into this flexible RKHS function class the possibility to impose *hard* linear shape requirements on derivatives, i.e. constraints of the form

$$0 \leq b + Df(\mathbf{x}) \quad \forall \mathbf{x} \in K \tag{1}$$

for a bias $b \in \mathbb{R}$, given a differential operator $D = \sum_j \gamma_j \partial^{\mathbf{r}_j}$ where $\partial^{\mathbf{r}} f(\mathbf{x}) = \frac{\partial^{\sum_{j=1}^d r_j} f(\mathbf{x})}{\partial^{r_1}_{x_1} \cdots \partial^{r_d}_{x_d}}$ and a compact set $K \subset \mathbb{R}^d$. The fundamental technical challenge is to guarantee the pointwise inequality (1) at *all* points of the (typically non-finite) set $K$. We show that one can tighten the infinite number of affine constraints (1) into a finite number of second-order cone constraints

$$\eta \|f\| \leq b + Df(\mathbf{x}_m) \quad \forall m \in \{1, \ldots, M\} \tag{2}$$

for a suitable choice of $\eta > 0$ and $\{\mathbf{x}_m\}_{m=1\ldots M} \subseteq K$.

Our **contributions** can be summarized as follows.

1. We show that hard shape requirements can be embedded in kernel machines by taking a second-order cone (SOC) tightening of constraint (1), which can be readily tackled by convex solvers. Our formulation builds upon the reproducing property for kernel derivatives and on coverings of compact sets.

2. We prove error bounds on the distance between the solutions of the strengthened and original problems.

3. We achieve state-of-the-art performance in joint quantile regression (JQR) in RKHSs. We also combine JQR with other shape constraints in economics and in the analysis of aircraft trajectories.

The paper is structured as follows. Section 2 is about problem formulation. Our main result is presented in Section 3. Numerical illustrations are given in Section 4. Proofs and additional examples are provided in the supplement.

## 2  Problem formulation

In this section we formulate our problem after introducing some notations, which the reader may skip at first, and return to if necessary.

**Notations:** Let $\mathbb{N} := \{0, 1, \dots\}$, $\mathbb{N}^* := \{1, 2, \dots\}$ and $\mathbb{R}_+$ denote the set of natural numbers, positive integers and non-negative real numbers, respectively. We use the shorthand $[n] := \{1, \dots, n\}$. The $p$-norm of a vector $\mathbf{v} \in \mathbb{R}^p$ is $\|\mathbf{v}\|_p = \left(\sum_{j\in[d]} |v_j|^p\right)^{\frac{1}{p}}$ ($1 \le p < \infty$); $\|\mathbf{v}\|_\infty = \max_{j\in[d]} |v_j|$. The $j$-th canonical basis vector is $\mathbf{e}_j$; $\mathbf{0}_d \in \mathbb{R}^d$ is the zero vector. Let $\mathbb{B}_{\|\cdot\|}(\mathbf{c}, r) = \{\mathbf{x} \in \mathbb{R}^d : \|\mathbf{x} - \mathbf{c}\| \le r\}$ be the closed ball in $\mathbb{R}^d$ with center $\mathbf{c}$ and radius $r$ in norm $\|\cdot\|$. Given a norm $\|\cdot\|$ and radius $\delta > 0$, a $\delta$-net of a compact set $K \subset \mathbb{R}^d$ consists of a set of points $\{\mathbf{x}_m\}_{m\in[M]}$ such that $K \subseteq \cup_{m\in[M]} \mathbb{B}_{\|\cdot\|}(\mathbf{x}_m, \delta)$, in other words $\left\{\mathbb{B}_{\|\cdot\|}(\mathbf{x}_m, \delta)\right\}_{m\in[M]}$ forms a covering of $K$. The identity matrix is $\mathbf{I}$. For a matrix $\mathbf{M} \in \mathbb{R}^{d_1 \times d_2}$, $\mathbf{M}^\top \in \mathbb{R}^{d_2 \times d_1}$ denotes its transpose, its operator norm is $\|\mathbf{M}\| = \sup_{\mathbf{x}\in\mathbb{R}^{d_2}:\|\mathbf{x}\|_2=1} \|\mathbf{M}\mathbf{x}\|_2$. The inverse of a non-singular matrix $\mathbf{M} \in \mathbb{R}^{d\times d}$ is $\mathbf{M}^{-1} \in \mathbb{R}^{d\times d}$. The concatenation of matrices $\mathbf{M}_1 \in \mathbb{R}^{d_1 \times d}, \dots, \mathbf{M}_N \in \mathbb{R}^{d_N \times d}$ is denoted by $\mathbf{M} = [\mathbf{M}_1; \dots; \mathbf{M}_N] \in \mathbb{R}^{\left(\sum_{n\in[N]} d_n\right)\times d}$. Let $\mathfrak{X}$ be an open subset of $\mathbb{R}^d$ with a real-valued kernel $k : \mathfrak{X} \times \mathfrak{X} \to \mathbb{R}$, and associated reproducing kernel Hilbert space (RKHS) $\mathcal{F}_k$. The Hilbert space $\mathcal{F}_k$ is characterized by $f(\mathbf{x}) = \langle f, k(\mathbf{x}, \cdot)\rangle_k$ ($\forall \mathbf{x} \in \mathfrak{X}, \forall f \in \mathcal{F}_k$) and $k(\mathbf{x}, \cdot) \in \mathcal{F}_k$ ($\forall \mathbf{x} \in \mathfrak{X}$) where $\langle \cdot, \cdot\rangle_k$ stands for the inner product in $\mathcal{F}_k$, and $k(\mathbf{x}, \cdot)$ denotes the function $\mathbf{y} \in \mathfrak{X} \mapsto k(\mathbf{x}, \mathbf{y}) \in \mathbb{R}$. The first property is called the reproducing property, the second one describes a generating family of $\mathcal{F}_k$. The norm on $\mathcal{F}_k$ is written as $\|\cdot\|_k$. For a multi-index $\mathbf{r} \in \mathbb{N}^d$ let the $\mathbf{r}$-th order partial derivative of a function $f$ be denoted by $\partial^{\mathbf{r}} f(\mathbf{x}) = \frac{\partial^{|\mathbf{r}|} f(\mathbf{x})}{\partial x_1^{r_1} \cdots \partial x_d^{r_d}}$ where $|\mathbf{r}| = \sum_{j\in[d]} r_j$ is the length of $\mathbf{r}$. When $d = 1$ the $f^{(n)} = \partial^n f$ notation is applied; specifically $f''$ and $f'$ are used in case of $n = 2$ and $n = 1$. Given $s \in \mathbb{N}$, let $\mathcal{C}^s(\mathfrak{X})$ be the set of real-valued functions on $\mathfrak{X}$ with continuous derivatives up to order $s$ (i.e., $\partial^{\mathbf{r}} f \in \mathcal{C}(\mathfrak{X}) := \mathcal{C}^0(\mathfrak{X})$ when $|\mathbf{r}| \le s$). Let $I \in \mathbb{N}^*$. Given $(A_i)_{i\in[I]}$ sets let $\prod_{i\in[I]} A_i$ denote their Cartesian product; we write $A^I$ in case of $A = A_1 = \dots = A_I$.

Our **goal** is to solve hard shape-constrained kernel machines of the form

$$\left(\bar{\mathbf{f}}, \bar{\boldsymbol{b}}\right) = \underset{\mathbf{f}=(f_q)_{q\in[Q]} \in (\mathcal{F}_k)^Q,\, \mathbf{b}=(b_p)_{p\in[P]} \in \mathcal{B},\, (\mathbf{f}, \mathbf{b}) \in C}{\arg\min} \mathcal{L}(\mathbf{f}, \boldsymbol{b}), \tag{$\mathcal{P}$}$$

where we are given an objective function $\mathcal{L}$ and a constraint set $C$ (detailed below), a closed convex constraint set $\mathcal{B} \subseteq \mathbb{R}^P$ on the biases, an order $s \in \mathbb{N}$, an open set $\mathfrak{X} \subseteq \mathbb{R}^d$ with a kernel $k \in \mathcal{C}^s(\mathfrak{X}\times\mathfrak{X})$ and associated RKHS $\mathcal{F}_k$, and samples $S = \{(\mathbf{x}_n, y_n)\}_{n\in[N]} \subset \mathfrak{X} \times \mathbb{R}$. The objective function in $(\mathcal{P})$ is specified by the triplet $(S, L, \Omega)$:

$$\mathcal{L}(\mathbf{f}, \boldsymbol{b}) = L\left(\mathbf{b}, \left(\mathbf{x}_n, y_n, (f_q(\mathbf{x}_n))_{q\in[Q]}\right)_{n\in[N]}\right) + \Omega\left((\|f_q\|_k)_{q\in[Q]}\right),$$

with loss function $L : \mathbb{R}^P \times \left(\mathfrak{X} \times \mathbb{R} \times \mathbb{R}^Q\right)^N \to \mathbb{R}$ and regularizer $\Omega : (\mathbb{R}_+)^Q \to \mathbb{R}$. Notice that the objective $\mathcal{L}$ depends on the samples $S$ which are assumed to be fixed, hence our proposed optimization framework focuses on the empirical risk. The bias $\mathbf{b} \in \mathbb{R}^P$ can be both constraint (such as $f(x) \ge b_1$, $f'(x) \ge b_2$) and variable-related ($f_q + b_q$, see (4)-(5) later). The restriction of $L$ to $\mathcal{B}$ is assumed to be strictly convex in $\mathbf{b}$, and $\Omega$ is supposed to be strictly increasing in each of its arguments to ensure the uniqueness of minimizers of $(\mathcal{P})$.

The $I \in \mathbb{N}^*$ hard shape requirements in $(\mathcal{P})$ take the form[1]

$$C = \left\{(\mathbf{f}, \mathbf{b}) \,|\, (\mathbf{b}_0 - \mathbf{U}\mathbf{b})_i \le D_i(\mathbf{W}\mathbf{f} - \mathbf{f}_0)_i(\mathbf{x}), \forall \mathbf{x} \in K_i, \forall i \in [I]\right\}, \tag{$\mathcal{C}$}$$

i.e., ($\mathcal{C}$) encodes affine constraints of at most $s$-order derivatives ($D_i = \sum_{j \in [n_{i,j}]} \gamma_{i,j} \partial^{\mathbf{r}_{i,j}}$, $|\mathbf{r}_{i,j}| \leq s$, $\gamma_{i,j} \in \mathbb{R}$). Possible shifts are expressed by the terms $\mathbf{b}_0 = (b_{0,i})_{i \in [I]} \in \mathbb{R}^I$, $\mathbf{f}_0 = (f_{0,i})_{i \in [I]} \in (\mathcal{F}_k)^I$. The matrices $\mathbf{U} \in \mathbb{R}^{I \times P}$ and $\mathbf{W} \in \mathbb{R}^{I \times Q}$ capture the potential interactions within the bias variables $(b_p)_{p \in [P]}$ and functions $(f_q)_{q \in [Q]}$, respectively. The compact sets $K_i \subset \mathcal{X}$ ($i \in [I]$) define the domain where the constraints are imposed.

**Remarks:**

- Differential operators: As $\mathcal{X} \subseteq \mathbb{R}^d$ is open and $k \in \mathcal{C}^s(\mathcal{X} \times \mathcal{X})$, any differential operator $D_i$ of order at most $s$ is well defined (Saitoh and Sawano, 2016, Theorems 2.5 and 2.6, page 76) as a mapping from $\mathcal{F}_k$ to $\mathcal{C}(\mathcal{X})$. Since the coefficients $\{\gamma_{i,j}\}_{j \in [n_{i,j}]}$ of $D_i$-s belong to the whole $\mathbb{R}$, ($\mathcal{C}$) can cover inequality constraints in both directions.

- Bias constraint $\mathcal{B}$: Choosing $\mathcal{B} = \{\mathbf{0}_P\}$ leads to constant l.h.s. $\mathbf{b}_0$ in ($\mathcal{C}$). The other extreme is $\mathcal{B} = \mathbb{R}^P$ in which case no hard constraint is imposed on the bias variable $\mathbf{b}$.

- Compactness of $K_i$-s: The compactness assumption on the sets $K_i$ is exploited in the construction of our strengthened optimization problem (Section 3). This requirement also ensures not imposing restrictions "too far" from the observation points, which could be impossible to satisfy. Indeed, let us consider for instance a $c_0$-kernel $k$ on $\mathbb{R}$, i.e. that $k(x, \cdot) \in \mathcal{C}^0(\mathbb{R})$ for all $x$ and $\lim_{|y| \to \infty} k(x, y) = 0$ for all $x \in \mathbb{R}$ (as for the Gaussian kernel). In this case $\lim_{|y| \to \infty} f(y) = 0$ also holds for all $f \in \mathcal{F}_k$. Hence a constraint of the form "for all $t \in \mathbb{R}_+$, $f(t) \geq \epsilon > 0$" can *not* be satisfied for $f \in \mathcal{F}_k$.

- Assumption on $\mathcal{X}$: If $s = 0$ (in other words only function evaluations are present in the shape constraints), then $\mathcal{X}$ can be any topological space.

We give various **examples** for the considered problem family ($\mathcal{P}$). We start with an example where $Q = 1$.

**Kernel ridge regression** (KRR) with *monotonicity* constraint: In this case the objective function and constraint are

$$\mathcal{L}(f, b) := \frac{1}{N} \sum_{n \in [N]} |y_n - f(x_n)|^2 + \lambda_f \|f\|_k^2, \text{ s.t. } f'(x) \geq 0, \forall x \in [x_l, x_u] \qquad (3)$$

with $\lambda_f > 0$. In other words in ($\mathcal{P}$) we have $Q = 1$, $d = 1$, $s = 1$, $P = I = 1$, $K_1 = [x_l, x_u]$, $\Omega(z) = \lambda_f z^2$, $D_1 = \partial^1$, $U = W = 1$, $f_{1,0} = 0$, $b_{1,0} = 0$, and $b \in \mathcal{B} = \{0\}$ is a dummy variable.

**Joint quantile regression** (JQR; e.g. Sangnier et al., 2016): Given $0 < \tau_1 < \ldots < \tau_Q < 1$ levels the goal is to estimate *jointly* the $(\tau_1, \ldots, \tau_Q)$-quantiles of the conditional distribution $\mathbb{P}(Y|X = \mathbf{x})$ where $Y$ is real-valued. In this case the objective function is

$$\mathcal{L}(\mathbf{f}, \mathbf{b}) = \frac{1}{N} \sum_{q \in [Q]} \sum_{n \in [N]} l_{\tau_q} (y_n - [f_q(\mathbf{x}_n) + b_q]) + \lambda_{\mathbf{b}} \|\mathbf{b}\|_2^2 + \lambda_f \sum_{q \in [Q]} \|f_q\|_k^2, \qquad (4)$$

where $\lambda_{\mathbf{b}} > 0$, $\lambda_f > 0$,[2] and the pinball loss is defined as $l_\tau(e) = \max(\tau e, (\tau - 1)e)$ with $\tau \in (0, 1)$. In JQR, the estimated $\tau_q$-quantile functions $\{f_q + b_q\}_{q \in [Q]}$ are *not* independent; the joint shape constraint they should satisfy is the monotonically increasing property w.r.t. the quantile level $\tau$. It is natural to impose this *non-crossing* requirement on the smallest rectangle containing the points $\{\mathbf{x}_n\}_{n \in [N]}$, i.e. $K = \prod_{r \in [d]} \left[ \min\{(\mathbf{x}_n)_r\}_{n \in [N]}, \max\{(\mathbf{x}_n)_r\}_{n \in [N]} \right]$. The corresponding shape constraint is

$$f_{q+1}(\mathbf{x}) + b_{q+1} \geq f_q(\mathbf{x}) + b_q, \forall q \in [Q-1], \forall \mathbf{x} \in K. \qquad (5)$$

One gets (4)-(5) from ($\mathcal{P}$) by choosing $P = Q$, $I = Q - 1$, $s = 0$, $\mathbf{b}_0 = \mathbf{0}$, $\mathbf{f}_0 = \mathbf{0}$, $\mathcal{B} = \mathbb{R}^P$,

$$K_i = K \ (\forall i \in [I]), \Omega(\mathbf{z}) = \lambda_f \sum_{q \in [Q]} (z_q)^2, \text{ and } \mathbf{U} = \mathbf{W} = \begin{bmatrix} -1 & 1 & 0 & 0 \\ 0 & -1 & 1 & 0 \\ \vdots & \ddots & \ddots & \ddots \\ 0 & 0 & -1 & 1 \end{bmatrix} \in \mathbb{R}^{(Q-1) \times Q}.$$

**Further examples**: There are various other widely-used shape constraints beyond non-negativity (for which $s = 0$), monotonicity ($s = 1$) or convexity ($s = 2$) which can be taken into account in ($\mathcal{C}$). For instance one can consider $n$-monotonicity ($s = n$), $(n - 1)$-alternating monotonicity,

monotonicity w.r.t. unordered weak majorization ($s = 1$) or w.r.t. product ordering ($s = 1$), or supermodularity ($s = 2$). For details on how these shape constraints can be written as $(\mathcal{C})$, see the supplement (Section C).

## 3 Results

In this section, we first present our strengthened SOC-constrained problem, followed by a representer theorem and explicit bounds on the distance to the solution of $(\mathcal{P})$.

In order to introduce our proposed tightening, let us first consider the discretization of the $I$ constraints using $M_i$ points $\{\tilde{\mathbf{x}}_{i,m}\}_{m \in [M_i]} \subseteq K_i$. This would lead to the following relaxation of $(\mathcal{P})$

$$v_{\text{disc}} = \min_{\mathbf{f} \in (\mathcal{F}_k)^Q, \, \boldsymbol{b} \in \mathcal{B}} \mathcal{L}(\mathbf{f}, \boldsymbol{b}) \text{ s.t. } (\mathbf{b}_0 - \mathbf{Ub})_i \leq \min_{m \in [M_i]} D_i(\mathbf{Wf} - \mathbf{f}_0)_i \, (\tilde{\mathbf{x}}_{i,m}) \, \forall i \in [I]. \quad (6)$$

Our proposed SOC-constrained tightening can be thought of as adding extra, appropriately chosen, $\eta_i$-buffers to the l.h.s. of the constraints:

$$(\mathbf{f}_{\boldsymbol{\eta}}, \mathbf{b}_{\boldsymbol{\eta}}) = \underset{\mathbf{f} \in (\mathcal{F}_k)^Q, \, \boldsymbol{b} \in \mathcal{B} \subset \mathbb{R}^p}{\arg \min} \mathcal{L}(\mathbf{f}, \boldsymbol{b}) \quad (\mathcal{P}_{\boldsymbol{\eta}})$$

s.t.

$$(\mathbf{b}_0 - \mathbf{Ub})_i + \eta_i \| (\mathbf{Wf} - \mathbf{f}_0)_i \|_k \leq \min_{m \in [M_i]} D_i(\mathbf{Wf} - \mathbf{f}_0)_i \, (\tilde{\mathbf{x}}_{i,m}) \, \forall i \in [I]. \quad (\mathcal{C}_{\boldsymbol{\eta}})$$

The SOC constraint $(\mathcal{C}_{\boldsymbol{\eta}})$ is determined by a fixed $\boldsymbol{\eta} = (\eta_i)_{i \in [I]} \in \mathbb{R}_+^I$ coefficient vector and by the points $\{\tilde{\mathbf{x}}_{i,m}\}$.[3] For each $i \in [I]$, the points $\{\tilde{\mathbf{x}}_{i,m}\}_{m \in [M_i]}$ are chosen to form a $\delta_i$-net of the compact set $K_i$ for some $\delta_i > 0$ and a pre-specified norm $\|\cdot\|_{\mathcal{X}}$.[4] Given $\{\tilde{\mathbf{x}}_{i,m}\}_{m \in [M_i]}$, the coefficients $\eta_i \in \mathbb{R}_+$ are then defined as

$$\eta_i = \sup_{m \in [M_i], \, \mathbf{u} \in \mathbb{B}_{\|\cdot\|_{\mathcal{X}}}(\mathbf{0}, 1)} \| D_{i, \mathbf{x}} k(\tilde{\mathbf{x}}_{i,m}, \cdot) - D_{i, \mathbf{x}} k(\tilde{\mathbf{x}}_{i,m} + \delta_i \mathbf{u}, \cdot) \|_k, \quad (8)$$

where $D_{i, \mathbf{x}} k(\mathbf{x}_0, \cdot)$ is a shorthand for $\mathbf{y} \mapsto D_i(\mathbf{x} \mapsto k(\mathbf{x}, \mathbf{y}))(\mathbf{x}_0)$. Problem $(\mathcal{P}_{\boldsymbol{\eta}})$ has $I$ scalar SOC constraints $(\mathcal{C}_{\boldsymbol{\eta}})$ over infinite-dimensional variables. Let $\bar{v} = \mathcal{L}(\bar{\mathbf{f}}, \bar{\boldsymbol{b}})$ be the minimal value of $(\mathcal{P})$ and $v_{\boldsymbol{\eta}} = \mathcal{L}(\mathbf{f}_{\boldsymbol{\eta}}, \mathbf{b}_{\boldsymbol{\eta}})$ be that of $(\mathcal{P}_{\boldsymbol{\eta}})$. Notice that, when formally setting $\boldsymbol{\eta} = \mathbf{0}$, $(\mathcal{P}_{\boldsymbol{\eta}})$ corresponds to (6).

In our main result below (i) shows that $(\mathcal{C}_{\boldsymbol{\eta}})$ is indeed a tightening of $(\mathcal{C})$, (ii) provides a representer theorem which allows to solve numerically $(\mathcal{P}_{\boldsymbol{\eta}})$, and (iii) gives bounds on the difference between the solution of $(\mathcal{P}_{\boldsymbol{\eta}})$ and that of $(\mathcal{P})$ as a function of $(v_{\boldsymbol{\eta}} - v_{\text{disc}})$ and $\boldsymbol{\eta}$ respectively.

**Theorem** (Tightened task, representer theorem, bounds). *Let* $\mathbf{f}_{\boldsymbol{\eta}} = (f_{\boldsymbol{\eta}, q})_{q \in [Q]}$. *Then,*

*(i) Tightening: any* $(\mathbf{f}, \boldsymbol{b})$ *satisfying* $(\mathcal{C}_{\boldsymbol{\eta}})$ *also satisfies* $(\mathcal{C})$, *hence* $v_{disc} \leq \bar{v} \leq v_{\boldsymbol{\eta}}$.

*(ii) Representer theorem: For all* $q \in [Q]$, *there exist real coefficients* $\tilde{a}_{i,0,q}, \tilde{a}_{i,m,q}, a_{n,q} \in \mathbb{R}$ *such that* $f_{\boldsymbol{\eta}, q} = \sum_{i \in [I]} \left[ \tilde{a}_{i,0,q} f_{0,i} + \sum_{m \in [M_i]} \tilde{a}_{i,m,q} D_{i, \mathbf{x}} k(\tilde{\mathbf{x}}_{i,m}, \cdot) \right] + \sum_{n \in [N]} a_{n,q} k(\mathbf{x}_n, \cdot)$.

*(iii) Performance guarantee: if* $\mathcal{L}$ *is* $(\mu_{f_q}, \mu_{\boldsymbol{b}})$*-strongly convex w.r.t.* $(f_q, \mathbf{b})$ *for any* $q \in [Q]$, *then*

$$\| f_{\boldsymbol{\eta}, q} - \bar{f}_q \|_k \leq \sqrt{\frac{2(v_{\boldsymbol{\eta}} - v_{disc})}{\mu_{f_q}}}, \qquad \| \boldsymbol{b}_{\boldsymbol{\eta}} - \bar{\boldsymbol{b}} \|_2 \leq \sqrt{\frac{2(v_{\boldsymbol{\eta}} - v_{disc})}{\mu_{\boldsymbol{b}}}}. \quad (9)$$

*If in addition* $\mathbf{U}$ *is of full row-rank (i.e. surjective),* $\mathcal{B} = \mathbb{R}^P$, *and* $\mathcal{L}(\bar{\mathbf{f}}, \cdot)$ *is* $L_b$*-Lipschitz continuous on* $\mathbb{B}_{\|\cdot\|_2}(\bar{\mathbf{b}}, c_f \|\boldsymbol{\eta}\|_{\infty})$ *where* $c_f = \sqrt{I} \left\| (\mathbf{U}^\top \mathbf{U})^{-1} \mathbf{U}^\top \right\| \max_{i \in [I]} \left\| (\mathbf{Wf} - \mathbf{f}_0)_i \right\|_k$, *then*

$$\| f_{\boldsymbol{\eta}, q} - \bar{f}_q \|_k \leq \sqrt{\frac{2 L_b c_f \|\boldsymbol{\eta}\|_{\infty}}{\mu_{f_q}}}, \qquad \| \boldsymbol{b}_{\boldsymbol{\eta}} - \bar{\boldsymbol{b}} \|_2 \leq \sqrt{\frac{2 L_b c_f \|\boldsymbol{\eta}\|_{\infty}}{\mu_{\boldsymbol{b}}}}. \quad (10)$$

**Proof** (idea): The SOC-based reformulation relies on rewriting the constraint $(\mathcal{C})$ as the inclusion of the sets $\Phi_{D_i}(K_i)$ in the closed halfspaces $H^+_{\phi_i,\beta_i} := \{g \in \mathcal{F}_k \mid \langle \phi_i, g \rangle_k \geq \beta_i\}$ for $\forall i \in [I]$ where $\Phi_{D_i}(\mathbf{x}) := D_{i,\mathbf{x}}k(\mathbf{x},\cdot) \in \mathcal{F}_k$, $\Phi_{D_i}(X) := \{\Phi_{D_i}(\mathbf{x}) \mid \mathbf{x} \in X\}$, $\phi_i := (\mathbf{Wf} - \mathbf{f}_0)_i$ and $\beta_i := (\mathbf{b}_0 - \mathbf{Ub})_i$. The tightening is obtained by guaranteeing these inclusions with an $\eta_i$-net of $\Phi_{D_i}(K_i)$ containing the $\delta_i$-net of $K_i$ when pushed to $\mathcal{F}_k$. The bounds stem from classical inequalities for strongly convex objective functions. The proof details of (i)-(iii) are available in the supplement (Section A).

**Remarks**:

The **representer theorem** allows one to express $(\mathcal{P}_{\boldsymbol{\eta}})$ as a finite-dimensional SOC-constrained problem:

$$\min_{\substack{\mathbf{A} \in \mathbb{R}^{N \times Q}, \, \boldsymbol{b} \in \mathcal{B}, \\ \tilde{\mathbf{A}} \in \mathbb{R}^{N \times Q}, \, \tilde{\mathbf{A}}_0 \in \mathbb{R}^{I \times Q}}} \mathcal{L}(\mathbf{f},\boldsymbol{b}) \text{ s.t. } (\mathbf{b}_0 - \mathbf{Ub})_i + \eta_i \left\| \mathbf{G}^{1/2}\mathbf{g}_i \right\|_2 \leq \min_{m \in [M_i]} (\mathbf{G}_{D_i}\mathbf{g}_i)_{I+N+m} \; \forall i \in [I], \; \forall q \in [Q],$$

where $\tilde{\mathbf{e}}_i \in \mathbb{R}^I$ and $\mathbf{e}_i \in \mathbb{R}^{I+N+M}$ are the canonical basis vectors, $\mathbf{g}_i := \left[ \tilde{\mathbf{A}}_0; \mathbf{A}; \tilde{\mathbf{A}} \right] \mathbf{W}^\top \tilde{\mathbf{e}}_i - \mathbf{e}_i$ and the coefficients of the components of $\mathbf{f}$ were collected to $\tilde{\mathbf{A}}_0 = [\tilde{a}_{i,0,q}]_{i \in [I], \, q \in [Q]} \in \mathbb{R}^{I \times Q}$, $\mathbf{A} = [a_{n,q}]_{n \in [N], \, q \in [Q]} \in \mathbb{R}^{N \times Q}$, $\tilde{\mathbf{A}} = [\tilde{\mathbf{a}}_{i,q}]_{i \in [I], \, q \in [Q]} \in \mathbb{R}^{M \times Q}$ with $M = \sum_{i \in [I]} M_i$ and $\tilde{\mathbf{a}}_{i,q} = [\tilde{a}_{i,m,q}]_{m \in [M_i]} \in \mathbb{R}^{M_i}$ ($i \in [I]$, $q \in [Q]$). In this finite-dimensional optimization task, $\mathbf{G} \in \mathbb{R}^{(I+N+M) \times (I+N+M)}$ is the Gram matrix of $(\{f_{0,i}\}_{i \in I}, \{k(\mathbf{x}_n, \cdot)\}_{n \in [N]}, \{D_{i,\mathbf{x}}k(\tilde{\mathbf{x}}_{i,m}, \cdot)\}_{m \in [M_i], i \in I})$, $\mathbf{G}_{D_i} \in \mathbb{R}^{(I+N+M) \times (I+N+M)}$ is the Gram matrix of the differentials $D_i$ of these functions, $\mathbf{G}^{1/2}$ is the matrix square root of the positive semi-definite $\mathbf{G}$.

The **bounds**[5] (9)-(10) show that smaller $\boldsymbol{\eta}$ gives tighter guarantee on the recovery of $\bar{\mathbf{f}}$ and $\bar{\mathbf{b}}$. Since $\left| \partial^{\mathbf{r}} f_{\boldsymbol{\eta},q}(\mathbf{x}) - \partial^{\mathbf{r}} \bar{f}_q(\mathbf{x}) \right| \leq \sqrt{\partial^{\mathbf{r},\mathbf{r}} k(\mathbf{x},\mathbf{x})} \left\| f_{\boldsymbol{\eta},q} - \bar{f}_q \right\|_k$ by the reproducing property and the Cauchy-Schwarz inequality, the bounds on $\| f_{\boldsymbol{\eta}} - \bar{f} \|_k$ can be propagated to pointwise bounds on the derivatives (for $|\mathbf{r}| \leq s$). We emphasize again that in our optimization problem $(\mathcal{P}_{\boldsymbol{\eta}})$ the samples $S = \{(\mathbf{x}_n, y_n)\}_{n \in [N]}$ are assumed to be fixed; in other words the bounds (9) and (10) are meant conditioned on $S$.

The **parameters** $M$, $\delta$ and $\boldsymbol{\eta}$ are strongly intertwined, their interplay reveals an *accuracy-computation tradeoff*. Consider a shift-invariant kernel ($k(\mathbf{x},\mathbf{y}) = k_0(\mathbf{x} - \mathbf{y})$, $\forall \mathbf{x}, \mathbf{y}$), then (8) simplifies to $\eta_i := \sup_{\mathbf{u} \in \mathbb{B}_{\|\cdot\|_{\mathcal{X}}}(\mathbf{0},1)} \sqrt{|2D_{i,\mathbf{x}}D_{i,\mathbf{y}}k_0(\mathbf{0}) - 2D_{i,\mathbf{x}}D_{i,\mathbf{y}}k_0(\delta_i \mathbf{u})|}$, where $D_{i,\mathbf{y}}$ is defined similarly to $D_{i,\mathbf{x}}$.[6] This expression of $\eta_i$ implies that whenever $D_{i,\mathbf{x}}D_{i,\mathbf{y}}k_0$ is $L_\delta$-Lipschitz[7] on $\mathbb{B}_{\|\cdot\|_{\mathcal{X}}}(\mathbf{0}, \delta_i)$, then $\eta_i \leq \sqrt{2L_\delta}\sqrt{\delta_i}$. By the previous point, a smaller $\boldsymbol{\eta}$ ensures a better recovery which can be guaranteed by smaller $\delta_i$-s, which themselves correspond to a larger number of centers ($M_i$-s) in the $\delta_i$-nets of the $K_i$-s. Hence, one can control the computational complexity by the total number $M$ of points in the nets. Indeed, most SOCP solvers rely on primal-dual interior point methods which have (in the worst-case) cubic complexity $\mathcal{O}\left((P + N + M)^3\right)$ per iterations (Alizadeh and Goldfarb, 2003). Controlling $M$ allows one to tackle hard shape-constrained problems in moderate dimensions ($d$); for details see Section 4. In practice, to reduce the number of coefficients in $f_{\eta,q}$, it is beneficial to recycle the points $\{\mathbf{x}_n\}_{n \in [N]}$ among the $M_i$ virtual centers, whenever the points belong to a constraint set $K_i$. This simple trick was possible in all our numerical examples and kept the computational expense quite benign. Supplement (Section B) presents an example of the actual computational complexity observed.

While in this work we focused on the optimization problem $(\mathcal{P})$ which contains *solely* infinite-dimensional SOC constraints $(\mathcal{C}_{\boldsymbol{\eta}})$, the proved $(\mathcal{C}_{\boldsymbol{\eta}}) \Rightarrow (\mathcal{C})$ implication can be of independent interest to tackle problems where other types of constraints are present.[8] For simplicity we formulated our

Table 1: Joint quantile regression on 9 UCI datasets. Compared techniques: Primal-Dual Coordinate Descent (PDCD, Sangnier et al., 2016) and the presented SOC technique. Rows: benchmarks. 2nd column: dimension ($d$). 3rd column: sample number ($N$). 4-5th columns: mean $\pm$ std of $100\times$value of the pinball loss for PDCD and SOC; smaller is better.

| Dataset | $d$ | $N$ | PDCD | SOC |
|---|---|---|---|---|
| engel | 1 | 235 | $48 \pm 8$ | $53 \pm 9$ |
| GAGurine | 1 | 314 | $61 \pm 7$ | $65 \pm 6$ |
| geyser | 1 | 299 | $105 \pm 7$ | $108 \pm 3$ |
| mcycle | 1 | 133 | $66 \pm 9$ | $62 \pm 5$ |
| ftcollinssnow | 1 | 93 | $154 \pm 16$ | $148 \pm 13$ |
| CobarOre | 2 | 38 | $159 \pm 24$ | $151 \pm 17$ |
| topo | 2 | 52 | $69 \pm 18$ | $62 \pm 14$ |
| caution | 2 | 100 | $88 \pm 17$ | $98 \pm 22$ |
| ufc | 3 | 372 | $81 \pm 4$ | $87 \pm 6$ |

result with uniform coverings ($\delta_i$, $\eta_i$, $i \in [I]$). However, we prove it for more general non-uniform coverings ($\delta_{i,m}$, $\eta_{i,m}$, $i \in [I]$, $m \in [M_i]$; see Section A). This can beneficial for sets with complex geometry (e.g. star-shaped) or when recyling of the samples was used to obtain coverings (as the samples in $S$ have no reason to be equally spaced); we provide an example (in economics) using a non-uniform covering in Section 4.

In practice, since the convergence speed of SOCP solvers depends highly on the condition number of $\mathbf{G}^{1/2}$, it is worth replacing $\mathbf{G}^{1/2}$ with $(\mathbf{G} + \epsilon_{\text{tol}}\mathbf{I})^{1/2}$, setting a tolerance $\epsilon_{\text{tol}} \simeq 10^{-4}$. As $G + \epsilon_{\text{tol}}\mathbf{I} \succcurlyeq G$ (in the sense of positive semi-definite matrices), this regularization strengthens further the SOC constraint. Moreover, SOCP modeling frameworks (e.g. CVXPY or CVXGEN) suggest to replace quadratic penalties (see (4)) with the equivalent $\sqrt{\sum_{q\in[Q]} \|f_q\|_k^2} \leq \tilde{\lambda}_f$ and $\|\boldsymbol{b}\|_2 \leq \tilde{\lambda}_{\boldsymbol{b}}$ forms. This stems from their reliance on internal primal-dual interior point techniques.

## 4 Numerical experiments

In this section we demonstrate the efficiency of the presented SOC technique to solve hard shape-constrained problems.[9] We focus on the task of joint quantile regression (JQR) where the conditional quantiles are encoded by the pinball loss (4) and the shape requirement to fulfill is the non-crossing property (5). Supplement (Section B) provides an additional illustration in kernel ridge regression (KRR, (3)) on the importance of enforcing hard shape constraints in case of increasing noise level.

- **Experiment-1**: We compare the performance of the proposed SOC technique on 9 UCI benchmark datasets with a state-of-the-art JQR solver relying on soft shape constraints.
- **Experiment-2**: We augment the non-crossing constraint of JQR with monotonicity and concavity. Our two examples here belong to economics and to the analysis of aircraft trajectories.

In our experiments we used a Gaussian kernel with bandwidth $\sigma$, ridge regularization parameter $\lambda_f$ and $\lambda_{\mathbf{b}}$ (or upper bounds $\tilde{\lambda}_f$ on $\sqrt{\sum_{q\in[Q]} \|f_q\|_k^2}$ and $\tilde{\lambda}_{\boldsymbol{b}}$ on $\|\boldsymbol{b}\|_2$). We learned jointly five quantile functions ($\tau_q \in \{0.1, 0.3, 0.5, 0.7, 0.9\}$). We used CVXGEN (Mattingley and Boyd, 2012) to solve ($\mathcal{P}_{\boldsymbol{\eta}}$); the experiments took from seconds to a few minutes to run on an i7-CPU 16GB-RAM laptop.

In our **first set of experiments** we compared the efficiency of the proposed SOC approach with the PDCD technique (Sangnier et al., 2016) which minimizes the same loss (4) but with a *soft* non-crossing encouraging regularizer. We considered 9 UCI benchmarks. Our datasets were selected with $d \in \{1, 2, 3\}$; to our best knowledge none of the available JQR solvers is able to guarantee in a hard fashion the non-crossing property of the learned quantiles out of samples even in this case. Each dataset was split into training ($70\%$) and test ($30\%$) sets; the split and the experiment were repeated twenty times. For each split, we optimized the hyperparameters ($\sigma, \tilde{\lambda}_f, \tilde{\lambda}_{\boldsymbol{b}}$) of SOC, searching over a

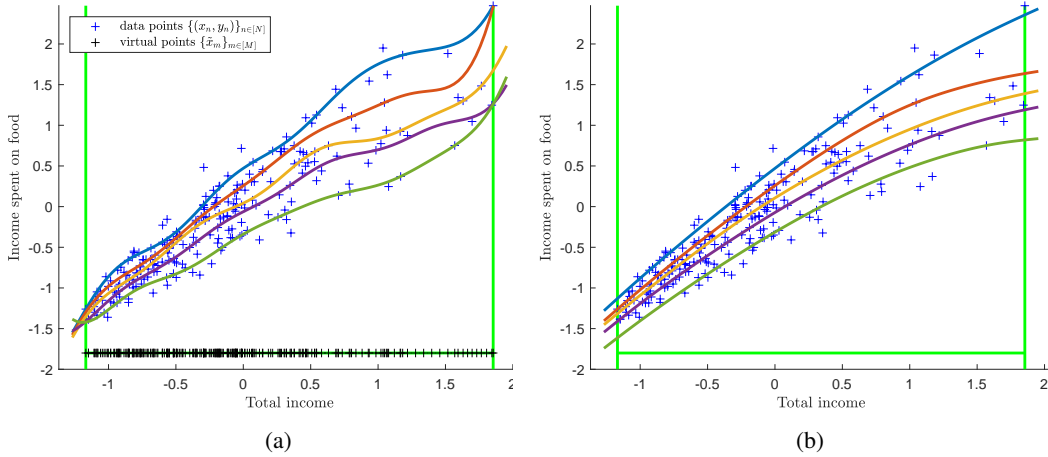

(a)                                          (b)

Figure 1: Joint quantile regression on the engel dataset using the SOC technique. Solid lines: estimated conditional quantile functions with $\tau_q \in \{0.1, 0.3, 0.5, 0.7, 0.9\}$ from bottom (dark green) to top (blue). Left plot: with non-crossing *and* increasing constraints. Right plot: with non-crossing, increasing *and* concavity constraints.

grid to minimize the pinball loss through a 5-fold cross validation on the training set. Particularly, the kernel bandwith $\sigma$ was searched over the square root of the deciles of the squared pairwise distance between the points $\{\mathbf{x}_n\}_{n \in [N]}$. The upper bound $\tilde{\lambda}_f$ on $\sqrt{\sum_{q \in [Q]} \|f_q\|_k^2}$ was scanned in the log-scale interval $[-1, 2]$. The upper bound $\tilde{\lambda}_{\boldsymbol{b}}$ on $\|\boldsymbol{b}\|_2$ was kept fixed: $\tilde{\lambda}_{\boldsymbol{b}} = 10 \max_{n \in [N]} |y_n|$. We then learned a model on the whole training set and evaluated it on the test set. The covering of $K = \prod_{r \in [d]} \left[ \min\{(\mathbf{x}_n)_r\}_{n \in [N]}, \max\{(\mathbf{x}_n)_r\}_{n \in [N]} \right]$ was carried out with $\| \cdot \|_2$-balls of radius $\delta$ chosen such that the number $M$ of added points was less than 100. This allowed for a rough covering while keeping the computation time for each run to be less than one minute. Our results are summarized in Table 1. The table shows that while the proposed SOC method guarantees the shape constraint in a *hard* fashion, its performance is on par with the state-of-the-art soft JQR solver.

In our **second set of experiments**, we demonstrate the efficiency of the proposed SOC estimator on tasks with additional hard shape constraints. Our first example is drawn from **economics**; we focused on JQR for the engel dataset, applying the same parameter optimization as in the first experiment. In this benchmark, the $\{(x_n, y_n)\}_{n \in [N]} \subset \mathbb{R}^2$ pairs correspond to annual household income ($x_n$) and food expenditure ($y_n$), preprocessed to have zero mean and unit variance. Engel's law postulates a monotone increasing property of $y$ w.r.t. $x$, as well as concavity. We therefore constrained the quantile functions to be non-crossing, monotonically increasing *and* concave. The interval $K = \left[ \min\{x_n\}_{n \in [N]}, \max\{x_n\}_{n \in [N]} \right]$ was covered with a non-uniform partition centered at the ordered centers $\{\tilde{x}_{m \in [M]}\}$ which included the original points $\{x_n\}_{n \in [N]}$ augmented with 15 virtual points. The radiuses were set to $\delta_{i,m} := \frac{\tilde{x}_{m+1} - \tilde{x}_m}{2}$ ($m \in [M-1]$, $i \in [I]$). The estimates with or without concavity are available in Fig. 1. It is interesting to notice that the estimated curves can intersect outside of the interval $K$ (see Fig. 1(a)), and that the additional concavity constraint mitigates this intersection (see Fig. 1(b)).

In our second example with extra shape constraints, we focused on the analysis of more than 300 **aircraft trajectories** (Nicol, 2013) which describe the radar-measured altitude ($y$) of aircrafts flying between two cities (Paris and Toulouse) as a function of time ($x$). These trajectories were restricted to their takeoff phase (where the monotone increasing property should hold), giving rise to a total number of samples $N = 15657$. We imposed non-crossing and monotonicity property. The resulting SOC-based quantile function estimates describing the aircraft takeoffs are depicted in Fig. 2. The plot illustrates how the estimated quantile functions respect the hard shape constraints and shows where the aircraft trajectories concentrate under various level of probability, defining a corridor of normal flight altitude values.

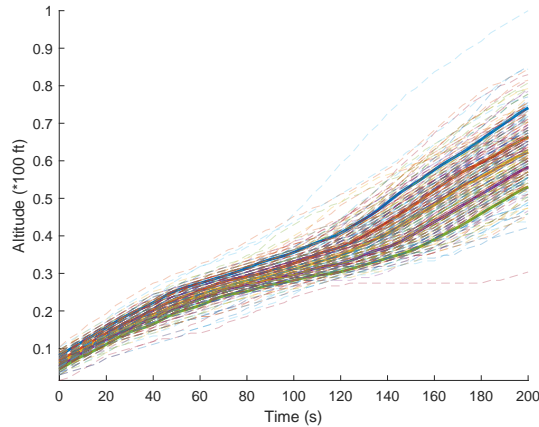

Figure 2: Joint quantile regression on aircraft takeoff trajectories. Number of samples: $N = 15657$. Shape constraints: non-crossing and increasing constraints. Dashed lines: trajectory samples. Solid lines: estimated conditional quantile functions.

These experiments demonstrate the efficiency of the proposed SOC-based solution to hard shape-constrained kernel machines.

## 5 Broader impact

Shape constraints play a central role in economics, social sciences, biology, finance, game theory, reinforcement learning and control problems as they enable more data-efficient computation and help interpretability. The proposed principled way of imposing hard shape constraints and algorithmic solution are expected to have positive impact in the aforementioned areas. For instance, from social perspective the studied quantile regression application can allow ensuring that safety regulations are better met. The improved sample efficiency, however, might result in dropping production indices and reduced privacy due to more target-specific applications.

## Acknowledgments and Disclosure of Funding

ZSz benefited from the support of the Europlace Institute of Finance and that of the Chair Stress Test, RISK Management and Financial Steering, led by the French École Polytechnique and its Foundation and sponsored by BNP Paribas.

## Footnotes

[1] In constraint $(\mathcal{C})$, $\mathbf{W}\mathbf{f}$ is meant as a formal matrix-vector product: $(\mathbf{W}\mathbf{f})_i = \sum_{q\in[Q]} W_{i,q} f_q$.

[2]Sangnier et al. (2016) considered the same loss but a *soft* non-crossing inducing regularizer inspired by matrix-valued kernels, and also set $\lambda_{\mathbf{b}} = 0$.

[3]Constraint $(\mathcal{C}_{\boldsymbol{\eta}})$ is the precise meaning of the preliminary intuition given in (2).

[4]The existence of finite $\delta_i$-nets ($M_i < \infty$) stems from the compactness of $K_i$-s. The flexibility in the choice of the norm $\|\cdot\|_{\mathcal{X}}$ allows for instance using cubes by taking the $\|\cdot\|_1$ or the $\|\cdot\|_{\infty}$-norm on $\mathbb{R}^d$ when covering the $K_i$-s.

[5] Notice that (9) is a computable bound, while (10) is not, as the latter depends on properties of the unknown solution of $(\mathcal{P})$.

[6] Similar computation can be carried out for higher order derivatives. For more general kernels, estimating $\eta_i$-s can be also done by sampling uniformly $\mathbf{u}$ in the unit ball.

[7] For instance any $C^{s+1}$ kernel satisfies this local Lipschitz requirement.

[8] For example having a unit integral is a natural additional requirement beyond non-negativity in density estimation, and writes as a linear equality constraint over the coefficients of $f_{\eta,q}$.

[9]The code replicating our numerical experiments is available at `https://github.com/PCAubin/Hard-Shape-Constraints-for-Kernels`.

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
