[Supplementary Material]

# Supplement

We provide the proof (Section A) of our main result presented in Section 3. Section B is about an additional numerical illustration in the context of kernel ridge regression on the importance of hard shape constraints in case of increasing level of noise. For completeness, reformulations of the additional shape constraint examples for ($\mathcal{C}$) mentioned at the end of Section 2 are detailed in Section C.

## A   Proof

For $i \in [I]$, we shall below denote $\phi_i = (\mathbf{Wf} - \mathbf{f}_0)_i$ and $\beta_i = (\mathbf{b}_0 - \mathbf{Ub})_i$. The proofs of the different parts are as follows.

(i) **Tightening**: By rewriting constraint ($\mathcal{C}$) using the derivative-reproducing property of kernels (Zhou, 2008; Saitoh and Sawano, 2016) we get

$$\langle \phi_i, D_{i,\mathbf{x}}k(\mathbf{x}, \cdot) \rangle_k = D_i \phi_i(\mathbf{x}) \geq \beta_i, \ \forall \mathbf{x} \in K_i. \tag{11}$$

Let us reformulate this constraint as an inclusion of sets

$$\Phi_{D_i}(K_i) \subseteq H^+_{\phi_i, \beta_i} := \{g \in \mathcal{F}_k \,|\, \langle \phi_i, g \rangle_k \geq \beta_i\},$$

where $\Phi_{D_i} : \mathbf{x} \mapsto D_{i,\mathbf{x}}k(\mathbf{x}, \cdot) \in \mathcal{F}_k$ and $\Phi_{D_i}(X) := \{\Phi_{D_i}(\mathbf{x}) \,|\, \mathbf{x} \in X\}$.

In order to get a finite geometrical description of $\Phi_{D_i}(K_i)$, we consider a finite covering of the compact set $K_i$:

$$\{\tilde{\mathbf{x}}_{i,m}\}_{m \in [M_i]} \subseteq K_i \subseteq \bigcup_{m \in [M_i]} \mathbb{B}_{\|\cdot\|_{\mathcal{X}}}\left(\tilde{\mathbf{x}}_{i,m}, \delta_{i,m}\right),$$

which implies that

$$\Phi_{D_i}(K_i) \subseteq \bigcup_{m \in [M_i]} \Phi_{D_i}\left(\mathbb{B}_{\|\cdot\|_{\mathcal{X}}}\left(\tilde{\mathbf{x}}_{i,m}, \delta_{i,m}\right)\right).$$

From the regularity of $k$, it follows that $\Phi_{D_i}$ is continuous from $\mathcal{X}$ to $\mathcal{F}_k$, and we define $\eta_{i,m} > 0$ ($i \in [I]$, $m \in [M_i]$) as

$$\eta_{i,m} := \sup_{\mathbf{u} \in \mathbb{B}_{\|\cdot\|_{\mathcal{X}}}(\mathbf{0},1)} \|\Phi_{D_i}\left(\tilde{\mathbf{x}}_{i,m}\right) - \Phi_{D_i}\left(\tilde{\mathbf{x}}_{i,m} + \delta_{i,m}\mathbf{u}\right)\|_k. \tag{12}$$

This means that for all $m \in [M_i]$

$$\Phi_{D_i}\left(\mathbb{B}_{\|\cdot\|_{\mathcal{X}}}\left(\tilde{\mathbf{x}}_{i,m}, \delta_{i,m}\right)\right) \subseteq \Phi_{D_i}\left(\tilde{\mathbf{x}}_{i,m}\right) + \mathbb{B}_k\left(0, \eta_{i,m}\right),$$

where $\mathbb{B}_k(0, \eta_{i,m}) := \{g \in \mathcal{F}_k \,|\, \|g\|_k \leq \eta_{i,m}\}$. In other words, for (11) to hold, it is sufficient that

$$\Phi_{D_i}\left(\tilde{\mathbf{x}}_{i,m}\right) + \mathbb{B}_k\left(0, \eta_{i,m}\right) \subseteq H^+_{\phi_i, \beta_i}, \ \forall m \in [M_i]. \tag{13}$$

By the definition of $H^+_{\phi_i, \beta_i}$, (13) is equivalent to

$$\beta_i \leq \inf_{g \in \mathbb{B}_k} \langle \phi_i, D_{i,\mathbf{x}}k(\tilde{\mathbf{x}}_{i,m}, \cdot) + \eta_{i,m}g \rangle_k = D_i(\phi_i)(\tilde{\mathbf{x}}_{i,m}) - \eta_{i,m}\|\phi_i\|_k, \ \forall m \in [M_i].$$

Taking the minimum over $m \in [M_i]$, we get

$$\|\phi_i\|_k \leq \min_{m \in [M_i]} \frac{1}{\eta_{i,m}} \left[-\beta_i + D_i(\phi_i)\left(\tilde{\mathbf{x}}_{i,m}\right)\right]. \tag{14}$$

Hence we proved that for any $(\mathbf{f}, \boldsymbol{b})$ satisfying (14), (11) also holds. The SOC-based reformulation is illustrated geometrically in Fig. 3. Constraint ($\mathcal{C}$) can be reformulated as requiring that the image $\Phi_{D_i}(K_i)$ of $K_i$ under the $D_i$-feature map $\Phi_{D_i}(\mathbf{x}) := D_{i,\mathbf{x}}k(\mathbf{x}, \cdot) \in \mathcal{F}_k$ is contained in the halfspace 'above' the affine hyperplane defined by normal vector $(\mathbf{Wf} - \mathbf{f}_0)_i$ and bias $(\mathbf{b}_0 - \mathbf{Ub})_i$. The discretization (6) of constraint ($\mathcal{C}$) at the points $\{\tilde{\mathbf{x}}_{i,m}\}_{m \in [M_i]}$ only requires the images $\Phi_{D_i}(\tilde{\mathbf{x}}_{i,m})$ of the points to be above the hyperplane. Constraint ($\mathcal{C}_{\boldsymbol{\eta}}$) instead inflates each of those points by a radius $\eta_i$.

Figure 3: Illustration of the SOC constraint $(\mathcal{C}_{\boldsymbol{\eta}})$.

(ii) **Representer theorem**: For any $q \in [Q]$, let $f_{\boldsymbol{\eta},q} = v_q + w_q$ where $v_q$ belongs to[10]

$$V := \operatorname{span}\left(\{f_{0,i}\}_{i \in I}, \{k(\mathbf{x}_n, \cdot)\}_{n \in [N]}, \{D_{i,\mathbf{x}}k(\tilde{\mathbf{x}}_{i,m}, \cdot)\}_{m \in [M_i], i \in [I]}\right) \subset \mathcal{F}_k$$

while $w_q$ is in the orthogonal complement of $V$ in $\mathcal{F}_k$ ($w_q \in V^{\perp} := \{w \in \mathcal{F}_k : \langle w, v \rangle_k = 0, \ \forall v \in V\}$). Let $\mathbf{v} := (v_q)_{q \in [Q]} \in (\mathcal{F}_k)^Q$. As constraint $(\mathcal{C}_{\boldsymbol{\eta}})$ holds for $(\mathbf{f}_{\boldsymbol{\eta}}, \boldsymbol{b}_{\boldsymbol{\eta}})$,

$$(\mathbf{b}_0 - \mathbf{U}\mathbf{b}_{\boldsymbol{\eta}})_i + \eta_i \|(\mathbf{W}\mathbf{f}_{\boldsymbol{\eta}} - \mathbf{f}_0)_i\|_k \leq \min_{m \in [M_i]} D_i(\mathbf{W}\mathbf{f}_{\boldsymbol{\eta}} - \mathbf{f}_0)_i \left(\tilde{\mathbf{x}}_{i,m}\right), \forall i \in [I].$$

However $(\mathbf{v}, \boldsymbol{b}_{\boldsymbol{\eta}})$ also satisfies $(\mathcal{C}_{\boldsymbol{\eta}})$ since $\|(\mathbf{W}\mathbf{v} - \mathbf{f}_0)_i\|_k \leq \|(\mathbf{W}\mathbf{f}_{\boldsymbol{\eta}} - \mathbf{f}_0)_i\|_k$ and $D_i(\mathbf{W}\mathbf{v} - \mathbf{f}_0)_i (\tilde{\mathbf{x}}_{i,m}) = D_i(\mathbf{W}\mathbf{f}_{\boldsymbol{\eta}} - \mathbf{f}_0)_i (\tilde{\mathbf{x}}_{i,m})$:

$$\left\|(\mathbf{W}\mathbf{f}_{\boldsymbol{\eta}} - \mathbf{f}_0)_i\right\|_k^2 = \left\|\sum_{q \in [Q]} W_{i,q} \underbrace{f_{\boldsymbol{\eta},q}}_{v_q + w_q} - f_{0,i}\right\|_k^2 = \left\|\underbrace{\sum_{q \in [Q]} W_{i,q} v_q - f_{0,i}}_{\in V} + \underbrace{\sum_{q \in [Q]} W_{i,q} w_q}_{\in V^{\perp}}\right\|_k^2$$

$$= \left\|\sum_{q \in [Q]} W_{i,q} v_q - f_{0,i}\right\|_k^2 + \left\|\sum_{q \in [Q]} W_{i,q} w_q\right\|_k^2 \geq \left\|\sum_{q \in [Q]} W_{i,q} v_q - f_{0,i}\right\|_k^2$$

$$= \|(\mathbf{W}\mathbf{v} - \mathbf{f}_0)_i\|_k^2,$$

$$D_i(\mathbf{W}\mathbf{f}_{\boldsymbol{\eta}} - \mathbf{f}_0)_i (\tilde{\mathbf{x}}_{i,m}) = D_i \left(\sum_{q \in [Q]} W_{i,q} \underbrace{f_{\boldsymbol{\eta},q}}_{v_q + w_q} - f_{0,i}\right) (\tilde{\mathbf{x}}_{i,m})$$

$$= D_i(\mathbf{W}\mathbf{v} - \mathbf{f}_0)_i (\tilde{\mathbf{x}}_{i,m}) + D_i \left(\sum_{q \in [Q]} W_{i,q} w_q\right) (\tilde{\mathbf{x}}_{i,m})$$

$$= D_i(\mathbf{W}\mathbf{v} - \mathbf{f}_0)_i (\tilde{\mathbf{x}}_{i,m}) + \underbrace{\left\langle \sum_{q \in [Q]} W_{i,q} w_q, D_{i,\mathbf{x}}k\left(\tilde{\mathbf{x}}_{i,m}, \cdot\right)\right\rangle_k}_{=0}$$

using the derivative-reproducing property of kernels, and that $\sum_{q \in [Q]} W_{i,q} w_q \in V^{\perp}$, while $D_{i,\mathbf{x}}k\left(\tilde{\mathbf{x}}_{i,m}, \cdot\right) \in V$. The regularizer $\Omega$ is assumed to be strictly increasing in each argument $\|f_{\boldsymbol{\eta},q}\|_k$. As $\|f_{\boldsymbol{\eta},q}\|_k^2 = \|v_q\|_k^2 + \|w_q\|_k^2$, and $(\mathbf{f}_{\boldsymbol{\eta}}, \mathbf{b}_{\boldsymbol{\eta}})$ minimizes $\mathcal{L}$, $w_q = 0$ necessarily; in other words $f_{\boldsymbol{\eta},q} \in V$ for all $q \in [Q]$.

(iii) **Performance guarantee**: From (i), we know that the solution $(\mathbf{f}_{\boldsymbol{\eta}}, \mathbf{b}_{\boldsymbol{\eta}})$ of $(\mathcal{P}_{\boldsymbol{\eta}})$ is also admissible for $(\mathcal{P})$. Discretizing the shape constraints is a relaxation of $(\mathcal{P})$. Hence $v_{\mathrm{disc}} \le \bar{v} \le v_{\boldsymbol{\eta}}$.

Let us fix any $(\mathbf{p}_f, \mathbf{p}_b) \in (\mathcal{F}_k)^Q \times \mathbb{R}^P$ belonging to the subdifferential of $\mathcal{L}(\cdot, \cdot) + \chi_{\mathcal{C}}(\cdot, \cdot)$ at point $(\bar{\mathbf{f}}, \bar{\mathbf{b}})$, where $\chi_{\mathcal{C}}$ is the characteristic function of $\mathcal{C}$, i.e. $\chi_{\mathcal{C}}(\mathbf{u}, \mathbf{v}) = 0$ if $(\mathbf{u}, \mathbf{v}) \in \mathcal{C}$ and $\chi_{\mathcal{C}}(\mathbf{u}, \mathbf{v}) = +\infty$ otherwise. Since $(\bar{\mathbf{f}}, \bar{\mathbf{b}})$ is the optimum of $(\mathcal{P})$, for any $(\mathbf{f}, \boldsymbol{b})$ admissible for $(\mathcal{P})$,

$$\sum_{q \in [Q]} \langle p_{f,q}, f_q - \bar{f}_q \rangle_k + \langle \mathbf{p}_b, \mathbf{b} - \bar{\mathbf{b}} \rangle_2 \ge 0, \tag{15}$$

where $\mathbf{p}_f = (p_{f,q})_{q \in [Q]}$. Hence using the $(\mu_{f_q}, \mu_{\mathbf{b}})$-strong convexity of $\mathcal{L}$ w.r.t. $(f_q, \mathbf{b})$ we get

$$\mathcal{L}(\mathbf{f}_{\boldsymbol{\eta}}, \boldsymbol{b}_{\boldsymbol{\eta}}) \ge \mathcal{L}(\bar{\mathbf{f}}, \bar{\boldsymbol{b}}) + \sum_{q \in [Q]} \langle p_{f_{\boldsymbol{\eta}}, q}, f_{\boldsymbol{\eta}, q} - \bar{f}_q \rangle_k + \langle \mathbf{p}_b, \boldsymbol{b}_{\boldsymbol{\eta}} - \bar{\boldsymbol{b}} \rangle_2 + \sum_{q \in [Q]} \frac{\mu_{f_q}}{2} \| f_{\boldsymbol{\eta}, q} - \bar{f}_q \|_k^2 \tag{16}$$

$$+ \frac{\mu_{\boldsymbol{b}}}{2} \| \boldsymbol{b}_{\boldsymbol{\eta}} - \bar{\boldsymbol{b}} \|_2^2.$$

As $v_{\boldsymbol{\eta}} - v_{\mathrm{disc}} \ge \mathcal{L}(\mathbf{f}_{\boldsymbol{\eta}}, \boldsymbol{b}_{\boldsymbol{\eta}}) - \mathcal{L}(\bar{\mathbf{f}}, \bar{\boldsymbol{b}})$, using the non-negativity (15) for $(\mathbf{f}_{\boldsymbol{\eta}}, \boldsymbol{b}_{\boldsymbol{\eta}})$, one gets from (16) the claimed bound (9).

To prove (10), recall that $(\bar{\mathbf{f}}, \bar{\mathbf{b}})$ satisfies $(\mathcal{C})$ and that we assume $\mathcal{B} = \mathbb{R}^P$. Let $\eta_i = \max_{m \in [M_i]} \eta_{i,m}$, $i \in [I]$ with $\eta_{i,m}$ defined in (12), and $\tilde{\mathbf{b}} = (\tilde{b}_i)_{i \in [I]} \in \mathbb{R}^I$ with

$$\tilde{b}_i := \eta_i \| (\mathbf{W}\bar{\mathbf{f}} - \mathbf{f}_0)_i \|_k. \tag{17}$$

As $\mathbf{U}$ is full-row rank, one can define its right inverse $(\mathbf{U}\mathbf{U}^+ = \mathbf{I})$ as $\mathbf{U}^+ = (\mathbf{U}^\top \mathbf{U})^{-1} \mathbf{U}^\top$. Then the pair $(\bar{\mathbf{f}}, \bar{\boldsymbol{b}} + \mathbf{U}^+ \tilde{\mathbf{b}})$ satisfies $(\mathcal{C}_{\boldsymbol{\eta}})$ since for any $m \in [M_i]$

$$\eta_i \| (\mathbf{W}\bar{\mathbf{f}} - \mathbf{f}_0)_i \|_k = \tilde{b}_i = (\mathbf{U}\mathbf{U}^+ \tilde{\mathbf{b}})_i \le (\mathbf{U}\mathbf{U}^+ \tilde{\mathbf{b}})_i + \underbrace{(\mathbf{U}\bar{\mathbf{b}} - \mathbf{b}_0)_i + D_i (\mathbf{W}\bar{\mathbf{f}} - \mathbf{f}_0)_i(\tilde{\mathbf{x}}_{i,m})}_{\ge 0}$$

$$= (\mathbf{U}(\bar{\mathbf{b}} + \mathbf{U}^+ \tilde{\mathbf{b}}) - \mathbf{b}_0)_i + D_i (\mathbf{W}\bar{\mathbf{f}} - \mathbf{f}_0)_i(\tilde{\mathbf{x}}_{i,m}).$$

Thus, $(\bar{\mathbf{f}}, \bar{\boldsymbol{b}} + \mathbf{U}^+ \tilde{\mathbf{b}})$ is admissible for $(\mathcal{P}_{\boldsymbol{\eta}})$ and as $(\mathbf{f}_{\boldsymbol{\eta}}, \boldsymbol{b}_{\boldsymbol{\eta}})$ is optimal for $(\mathcal{P}_{\boldsymbol{\eta}})$, we have

$$\mathcal{L}(\mathbf{f}_{\boldsymbol{\eta}}, \boldsymbol{b}_{\boldsymbol{\eta}}) - \mathcal{L}(\bar{\mathbf{f}}, \bar{\boldsymbol{b}}) \le \mathcal{L}(\bar{\mathbf{f}}, \bar{\boldsymbol{b}} + \mathbf{U}^+ \tilde{\mathbf{b}}) - \mathcal{L}(\bar{\mathbf{f}}, \bar{\boldsymbol{b}}) \overset{(a)}{\le} L_b \| \mathbf{U}^+ \tilde{\mathbf{b}} \|_2 \le L_b \| \mathbf{U}^+ \| \| \tilde{\mathbf{b}} \|_2$$

$$\le L_b \| \mathbf{U}^+ \| \sqrt{I} \| \tilde{\mathbf{b}} \|_\infty \overset{(b)}{\le} L_b \| \mathbf{U}^+ \| \| \boldsymbol{\eta} \|_\infty \sqrt{I} \max_{i \in [I]} \| (\mathbf{W}\bar{\mathbf{f}} - \mathbf{f}_0)_i \|_k$$

$$\overset{(c)}{=} L_b c_f \| \boldsymbol{\eta} \|_\infty,$$

where (a) stems from the local Lipschitz property of $\mathcal{L}$ $(\| \mathbf{U}^+ \tilde{\mathbf{b}} \|_2 \le c_f \| \boldsymbol{\eta} \|_\infty)$, (b) holds by (17), and (c) follows from the definition of $c_f$. Combined with (16), this gives the bound (10).

## B Shape-constrained kernel ridge regression

In this section we illustrate in kernel ridge regression (KRR, (3)) the importance of enforcing hard shape constraints in case of increasing noise level. We consider a synthetic dataset of $N = 30$ points from the graph of a quadratic function where the values $\{x_n\}_{n \in [N]} \subset \mathbb{R}$ were generated uniformly on $[-2, 2]$. The corresponding $y$-coordinates of the graph were perturbed by additive Gaussian noise:

$$y_n = x_n^2 + \epsilon_n \ (\forall n \in [N]), \{\epsilon_n\}_{n \in [N]} \overset{\text{i.i.d.}}{\sim} \mathcal{N}(0, \xi^2).$$

We impose a monotonically increasing shape constraint on the interval $[x_l, x_u] = [0, 2]$, and study the effect of the level of the added noise $(\xi)$ on the desired increasing property of the estimate without

(a)

(b)

(c)

(d)

Figure 4: (a): Illustration for kernel ridge regression. Observation: quadratic function perturbed with additive Gaussian noise. Shape constraint: monotone increasing property on $[0, 2]$. Compared techniques: regression without (KRR) and with hard shape constraint (SOC). (b): Violation of the shape constraint for the unconstrained KRR estimator as a function of the amplitude of the added Gaussian noise. Error measures: median of the proportion (left) and amount (right) of the violation of the monotone increasing property on $[0, 2]$. Dashed lines: lower and upper quartiles. (c): Evolution of the optimal objective values $v_{\boldsymbol{\eta}}$ and $v_{\text{disc}}$ when increasing the number $M$ of discretization points of the constraints on $[0, 2]$. (d): Computation time of $(\mathcal{P}_{\boldsymbol{\eta}})$ depending on the convex optimization solver (SeDuMi or MOSEK) selected.

(KRR) and with monotonic shape constraint (SOC). Here $\sigma = 0.5$ and $\lambda_f = 10^{-4}$, while $\xi$ varies in the interval $[0, 4]$.

Fig. 4(a) provides an illustration of the estimates in case of a fixed noise level $\xi = 1$. There is a good match between the KRR and SOC estimates outside of the interval $[0, 2]$, while the proposed SOC technique is able to correct the KRR estimate to enforce the monotonicity requirement on $[0, 2]$. In order to assess the performance of the unconstrained KRR estimator under varying level of noise, we repeated the experiment 1000 times for each noise level $\xi$ and computed the proportion and amount[11] of violation of the monotonicity requirement. Our results are summarized in Fig. 4(b). The figure shows that the error increases rapidly for KRR as a function of the noise level, and even for very low level of noise the monotonicity requirement does not hold. These experiments demonstrate that shape constraints can grossly be violated when facing noise if they are not enforced in an explicit and hard fashion. To illustrate the tightening property of Theorem 3, i.e. that $v_{\text{disc}} \leq \bar{v} \leq v_{\boldsymbol{\eta}}$, Fig. 4(c) shows the evolution of the optimal values $v_{\boldsymbol{\eta}}$ and $v_{\text{disc}}$ when increasing the number of discretization points ($M$) of the constraints on the constraint interval $[0, 2]$. Since by increasing $M$, we decrease $\eta$, the value $v_{\boldsymbol{\eta}}$ decreases, whereas $v_{\text{disc}}$ increases as the discretization incorporates more constraints. Larger value of $M$ naturally increases the polynomial computation time but not necessarily at the worst cubic expense, as shown in Fig. 4(d); the choice of the solver has also importance as it may provide a factor of two gain.

## C   Examples of handled shape constraints

In order to make the paper self-contained, in this section we provide the reformulations using derivatives of the additional shape constraints briefly mentioned at the end of Section 2: $n$-monotonicity ($s = n$; Chatterjee et al., 2015), $(n-1)$-alternating monotonicity (Fink, 1982), monotonicity w.r.t. unordered weak majorization ($s = 1$; Marshall et al., 2011, A.7. Theorem) or w.r.t. product ordering ($s = 1$), or supermodularity ($s = 2$; Simchi-Levi et al., 2014, Section 2).

Particularly, $n$-monotonicity ($n \in \mathbb{N}^*$) writes as $f^{(n)}(x) \geq 0$ ($\forall x$). $(n-1)$-alternating monotonicity[12] ($n \in \mathbb{N}^*$) is similar: for $n = 1$ non-negativity and non-increasing properties are required; for $n \geq 2$ $(-1)^j f^{(j)}$ has to be non-negative, non-increasing and convex for $\forall j \in \{0, \ldots, n-2\}$. The other examples are

- **Monotonicity w.r.t. partial ordering**: These generalized notions of monotonicity ($\mathbf{u} \preccurlyeq \mathbf{v} \Rightarrow f(\mathbf{u}) \leq f(\mathbf{v})$) rely on the partial orderings $\mathbf{u} \preccurlyeq \mathbf{v}$ iff $\sum_{j \in [i]} u_j \leq \sum_{j \in [i]} v_j$ for all $i \in [d]$ (unordered weak majorization) and $\mathbf{u} \preccurlyeq \mathbf{v}$ iff $u_i \leq v_i$ ($\forall i \in [d]$) (product ordering). For $C^1$ functions mononicity w.r.t. the unordered weak majorization is equivalent to

$$\partial^{\mathbf{e}_1} f(\mathbf{x}) \geq \ldots \geq \partial^{\mathbf{e}_d} f(\mathbf{x}) \geq 0 \quad (\forall \mathbf{x}).$$

Monotonicity w.r.t. product ordering for $C^1$ functions can be rephrased as

$$\partial^{\mathbf{e}_j} f(\mathbf{x}) \geq 0, \quad (\forall j \in [d], \quad \forall \mathbf{x}).$$

- **Supermodularity**: Supermodularity means that $f(\mathbf{u} \vee \mathbf{v}) + f(\mathbf{u} \wedge \mathbf{v}) \geq f(\mathbf{u}) + f(\mathbf{v})$ for all $\mathbf{u}, \mathbf{v} \in \mathbb{R}^d$, where maximum and minimum are meant coordinate-wise, i.e. $\mathbf{u} \vee \mathbf{v} := (\max(u_j, v_j))_{j \in [d]}$ and $\mathbf{u} \wedge \mathbf{v} := (\min(u_j, v_j))_{j \in [d]}$ for $\mathbf{u}, \mathbf{v} \in \mathbb{R}^d$. For $C^2$ functions this property corresponds to

$$\frac{\partial^2 f(\mathbf{x})}{\partial x_i \partial x_j} \geq 0 \quad (\forall i \neq j \in [d], \forall \mathbf{x}).$$

## Footnotes

[10]The linear hull of a finite set of points $(\mathbf{v}_m)_{m \in [M]}$ in a vector space is denoted by $\operatorname{span}(\{\mathbf{v}_m\}_{m \in [M]}) = \{\sum_{m \in [M]} a_m \mathbf{v}_m \mid a_m \in \mathbb{R}, \ \forall m \in [M]\}$.

[11]These performance measures are defined as $\frac{1}{2} \int_0^2 \max(0, -f'(x)) \mathrm{d}x$ and $\int_0^2 \max_{y \in [0, x]} [f(y) - f(x)] \mathrm{d}x$. By construction both measures are zero for SOC.

[12]For instance, the generator of a $d$-variate Archimedean copula can be characterized by $(d-2)$-alternating monotonicity (Malov, 2001; McNeil and Neslehová, 2009).