[Reviews · NeurIPS 2020]

Review 1

Summary and Contributions: This work aims to include shape constraints into kernel machines using an optimization framework. Specifically, the authors propose second-order cone optimization procedure using convex solvers.

Strengths: The paper addresses an important problem regarding imposing hard shape constraints - as opposed to typical soft constraints. The example regarding non-crossing joint quantile regression is particularly interesting.

Weaknesses: See Additional feedback.

Correctness: The general claims and method seem appropriate and correct.

Clarity: Could you clarify the meaning of the light green lines in the Figure 1 caption. It seems they refer to the interval boundaries.

Relation to Prior Work: See Additional feedback.

Reproducibility: Yes

Additional Feedback: Do you have any comments on how your hard shape constraint formulation affects overlapping quantiles versus the soft constraint (PDCD) ? It would be more informative to display Figure 1 alongside plots from alternative methods. Is it reasonable to attribute the comments on l304 regarding non-crossing to the additional regularizing properties of the concavity constraint? Regarding Table 1, it seems SOC performs _worse_ than PDCD on 5/9 datasets. This opens the question of when and where are hard constraints more beneficial over soft constraints. Furthermore, the question of computational complexity is not clear to me and what tradeoffs can be made when optimising for the enforced constraints. Following from the above, regarding the covering interval K and the comments on l296 that suggest we must sample virtual points within such an interval we want to enforce hard constraints on. How would such a scheme operate in higher dimensions - certainly it is not unreasonable to assume this sampling scheme would break in higher dimensions as the space needed to be covered grows. How would your hard constraint method perform compared to soft constraints in such a high dimensional regime? The exposition regarding the "virtual points M" is insufficient. They seem to play an important role in complexity however this is not discussed. What is the tradeoff with your methodology regarding M? There has been prior work for shape constrained Gaussian Processes [1] which are one of the more well known RKHS based modelling frameworks. Such work has not been mentioned - could you please remark on the connection between [1] and your work? [1] Solin, A., & Kok, M. (2019). Know Your Boundaries. In International Conference on Artificial Intelligence and Statistics. PMLR. ==== Given the author rebuttal I am increasing my score to Accept. In particular, I value the discussion on complexity and clarification with regard to relationship with GP work. I believe the direction of the work to be both interesting and important.


Review 2

Summary and Contributions: The paper addresses optimization with hard constraints in reproducing kernel Hilbert spaces (rkhs). The authors focus on linear constraints on derivatives of the target functions. Algorithmically, they propose a strategy based on a tightening of the constratings, that allows for a representer theorem and yields a practical algorithm. The main contributions are: - a practical algorithm to tackle such optimization problems; - a theoretical analysis relating the tightening coefficient with the original problem; - a thorough empirical evaluation of the method on a number of datasets.

Strengths: The idea of imposing hard differential constraints is relevant since it allows to tackle relevant problems in machine learning (density estimation, quantile regression, etc.) The proposed method is sound. The empirical evaluation is thorough. It makes the method appealing for the community.

Weaknesses: The authors give only examples of applications of their method to settings where 0-order differential constraints are imposed. It would be useful if they made concrete examples where additional orders (e.g. at least first-order constraints) need to be imposed. It would be beneficial to the paper if the authors discussed the computational aspects of the proposed approach, namely convergence rates and complexity. On this note, it would be interesting to extend the comparison of the proposed method with the state of the art also in terms of performance (e.g. time)

Correctness: To my understanding results are correct.

Clarity: Overall, the paper is well written and easy to understand. What it is not extremely evident is whether the tightening proposed by the authors is necessary to obtain the represented theorem (and thus the algorithm) and it could not be achieved by setting the tightening coefficient equal to 0. In the latter case, the authors should provide a thorough discussion better motivating the need for the tightening. In particular, in the introduction the concept of tightening appears suddenly in the list of contributions, but it is not properly introduced before. This affects the clarity of the paper.

Relation to Prior Work: yes

Reproducibility: Yes

Additional Feedback: ----- Post Authors' Response Phase ----- After reading the other reviews and the authors' response, I am convinced the paper to deserve acceptance: the work is rigorous, considers a relevant problem to the research community and proposes a novel method to address it. My recommendation for the authors would be to make a better case when introducing the problem, providing more examples of real-world problems that impose hard constraints (e.g. going into the details of some of the setting tackled in the experiments).


Review 3

Summary and Contributions: This paper proposes an approach to shape-constrained supervised learning problems, such as regression with a positivity constraint and quantile regression with a non-crossing constraint. The proposal is an optimization problem in an RKHS, with surrogate modeling of the shape-constraint using a finite set of points that cover the input region under consideration. Theoretical bounds are provided as a guarantee for the correctness of the proposed approach, with empirical validations on quantile regression.

Strengths: - The proposed approach is generic and applicable to a wide range of problems. - Theoretical bounds are provided, which show that the proposed approach is reasonable.

Weaknesses: - The proposed method is only applicable to problems with low dimensional input spaces. In fact, the largest dimensionality in the experiments is 3. This is due to the curse of dimensionality caused by the use of a finite set points that cover the region of interest in the surrogate modeling of the shape constraint. - The performance guarantee (Eq. 9) is not very significant, because it has not been shown how small the upper-bounds are. - The experiments are not intensive. The comparison was made only with the approach by Sangnier et al., 2016 on joint quantile regression.

Correctness: I haven't found any flaw in the paper, while I haven't check the proofs in the supplementary materials.

Clarity: - Strength: The writing is solid and mathematically rigorous. - Weakness: The paper is a bit dense and it's not easy to understand the contents intuitively.

Relation to Prior Work: I think so.

Reproducibility: Yes

Additional Feedback: - Computational costs of the proposed method should be discussed in detail, in terms of the relevant parameters. For instance, what is the computational complexity of solving Eq. (8)? - In lines 267-271, the authors said "to our best knowledge none of the available JQR solvers is able to guarantee in a hard fashion the non-crossing property of the learned quantiles out of samples even in this case". It would be nice if you can empirically demonstrate this. - In lines 286-288, the authors mention that `"The table shows that while the proposed SOC method guarantees the shape constraint in a hard fashion, its performance is on par with the state-of-the-art soft JQR solver." This is nice, but it would have been better if there are experimental results where the hard shape constraint results in significantly better performance. In the experiments, it would be interesting to see the effects of the choice of \delta (or M) ++++++ Post rebuttal ++++++++ Thanks for your feedback, which is convincing. So I increased my score.


Review 4

Summary and Contributions: This works proposes a new methodology for learning RKHS functions with shape constraints such as non-negativity, monotonicity, convexity etc. Using (finite) set covers (of compact sets) and the reproducing property, the idea is to come-up with stronger (SOC) constraints, which when hold at the core points, imply that the original constraints hold globally in the entire compact set. A formal bound on the apprximation as well as simulation results are provided.

Strengths: 1. I like how the simple and well-known observation that deviations in derivatives of functions can be upper bounded by those in the variable for RKHS functions, was used here to come-up with the tighter SOC constraint. Though such ideas are not fundamentally new in ML, I think it is a novelty in this line of work as far as I know. 2. Though there have been works that handle different kinds of constraints for e.g. non-negativity using sum-of-squares etc., this work seems to provide a unified framework for handling all such constraints. This is good. 3. Approximations bounds are provided (though standard, useful for the sake of completion).

Weaknesses: 1. My main complaint is the style of presentation and the flow of main ideas. For example, it would have been far more easier to understand and appreciate the work if the SOC was clearly derived in the main paper rather than postponing to the appendix. It would have also helped if a setting that is easier on notation is explained first and then the full-fledged generalization is presented. 2. Though some details are in lines 219-222, it would have been nice if more pointers/algos/special cases for sovling (8) are presented. Similarly a more detailed discussion on how to pick the core points \tilde{x}_{i,m} in (6) etc. would be of help. For example, how to pick the core points such that the bound in theorem would be low ? I imagine such details would make reading this more enjoyable. 3. One important comparison perhaps missing is: comparison with special case \eta_i=0 and even in this case, for increasing M_i. this may illustrate the importance of the SOC constraint.

Correctness: Seems ok. Please read (3) above.

Clarity: Not exactly. I took sometime to understand what is happening. Please see comments above.

Relation to Prior Work: this is kind of missing. For example, for each type of constraints like non-negativity, montonicity etc., the state-of-the-art (SOTA) is mentioned. e..g, sum-of-squares ideas for non-negativity etc. Then it is sadi that this work provides a unified framework that would compete well against these SOTA, then it would been better. However, I agree that at some level such a discussion is done in simulation section while comparing with PDCD.

Reproducibility: Yes

Additional Feedback: I have read the feedback and my queries were answered satisfactorily. So I am increasing the score.

[Author Response · NeurIPS 2020]

We thank the reviewers for their commitment and valuable insights despite the difficult times. We use Ri below to refer
to the $i^{th}$ reviewer. Questions/remarks are indicated by **Q** with reviewer identifiers in parentheses, answers are denoted
by **A**. To refer to line X in the submission we use the shorthand 'lX'. Our new figures are located on the r.h.s.

We briefly recall our **primary focus** (l55-58): to propose a flexible optimization framework capable of handling jointly
general hard shape constraints (expressible as affine inequalities over derivatives on compact sets) with rich function
classes (RKHSs). To the best of our knowledge, our approach is the first in this direction with guarantees. We are thus
less concerned about high-dimensional scalability questions, though we explicitly acknowledge it (l226) and provide
practical algorithms which allow a benign control of the computations in moderate dimension (l227-231, l258-259). We
note that specialized SOC solvers (instead of CVXGEN which we used for illustration) can provide additional speed-up.

**Q** (R1, R3): Table 1 (SOCP vs PDCD: comparable performance). R3: It would be nice to also demonstrate empirically
that JQR violates the imposed non-crossing constraints. **A**: We answer these 2 questions jointly. Following Sangnier et
al. 2016, a JQR method is considered to be favorable if (i) the technique gives comparable results in terms of pinball
loss (see our Table 1), and (ii) it violates the imposed shape constraints less often (SOCP respects it by construction,
whereas PDCD often produces crossings as it can be seen in the last column of Table 1 of Sangnier et al. 2016).

**Q** (R1, R3): higher dimensionality, soft shape constraint inducing regularizers.
**A**: In higher dimensions, compact coverings and our technique can still be
applied, though $\eta$ would be larger. Soft-constrained solutions (e.g. PDCD)
might run faster but again without guarantees.

**Q** (R1, R2): Role of virtual points (R1) and computational complexity (R2)
are not discussed. **A**: The complexity is $O((P+N+M)^3)$ in the worst case
(l226). As even a kernel ridge regression (KRR) scales cubically one can not
expect in general better behavior with additional hard shape constraints. We
provide the computational times for the reviewers associated to KRR with
monotonicity (Section B) on the r.h.s. In practice, recycling the $N$ sample
points among the $M$ virtual centers effectively reduces (l228-231, l259) the
number of coefficients to be determined (see $f_{\boldsymbol{\eta},q}$ in Theorem (ii)) and hence
the computational time.

**Q** (R1): Prior work for shape-constrained GPs could be added, like SK (A.
Solin & M. Kok., 2019). **A**: We cite from the GP literature C. Agrell (2019)
who handles shape constraints $a \leq \mathcal{L}f \leq b$ in GPs in a soft fashion where
$\mathcal{L}$ is a linear operator. SK tackles equality constraints ($f(\mathbf{x}) = 0$) on the
boundary of the domain of a GP in a hard way. Though SK's constraint
(equality of only function values on boundary) and its handling (computing
the eigendecomposition of the Laplace operator) are quite different from ours,
we are happy to refer to it for the sake of completeness.

**Q** (R2): The authors give only examples where 0-order differential constraints
are imposed. **A**: We consider higher order constraints in our examples in economics (Fig. 1(a): 0-1st order; Fig. 1(b):
0-1-2nd order), analysis of aircraft trajectories (Fig. 2, 0-1st order), and KRR with monotonicity (Fig. 4(a), 1st order).

**Q** (R2): Could a representer theorem be achieved by setting $\eta = 0$? **A**: Yes, however the choice of $\eta = 0$ would
correspond to the discretization (6) which does not ensure shape constraints in a hard way on the $K_i$-s.

**Q** (R3): Significance of (9)? **A**: (9) is a computable bound (footnote 5) and can be applied as an alternative stopping
criterion for the number of virtual points to add. In practice, we use the strategy detailed in l227-231 which works
reliably.

**Q** (R3, R4): How is (8) solved in practice? **A**: For instance in the JQR example (4)-(5), with a radial kernel
$k(\mathbf{x}, \mathbf{y}) = k_0(\|\mathbf{x} - \mathbf{y}\|_{\mathcal{X}})$ and $k_0$ monotonically decreasing (such as the Gaussian kernel) (8) simplifies (l220) to
$\eta_i = \sup_{\mathbf{u} \in \mathbb{B}_{\|\cdot\|_{\mathcal{X}}}(\mathbf{0},1)} \sqrt{|2k_0(0) - 2k_0(\delta_i \|\mathbf{u}\|_{\mathcal{X}})|} = \sqrt{|2k_0(0) - 2k_0(\delta_i)|}$; hence $\eta_i$ can be computed analytically.
Similar computation can be carried out for higher order derivatives. For more general kernels, estimating $\eta_i$-s can be
also done by sampling uniformly $\mathbf{u}$ in the unit ball.

**Q** (R3, R4): It would be interesting to see the effects of the choice of $\delta$ (or $M$) and compare it with $\eta = 0$. **A**: The
objective values for $\eta = 0$ are always below the optimal value of the original problem, while that of with $\eta > 0$ are
above. We provide an illustration (r.h.s.) that as $M$ is increasing the objective values get closer to each other.

**Q** (R4): How the virtual points $\mathbf{x}_{i,m}$-s are chosen in practice? **A**: According to our experiences, besides the recycling
trick (l227-231), choosing the $\mathbf{x}_{i,m}$-s to form an approximately uniform grid is a safe and reliable choice as it implies
uniformly small $\delta_i$, thus low bound on $\eta_i$ (l222), and hence tighter guarantee (see (10)).

We hope that we have answered all the questions of the reviewers.

[Meta-Review · NeurIPS 2020]

Four knowledgeable reviewers recommend accept, on the basis that this paper provides a universal approach to hard shape-constrained supervised learning that is applicable to a wide range of problems of interest to the NeurIPS community. I also recommend accept.